# Global Mental Health and Services for Migrants in Primary Care Settings in High-Income Countries: A Scoping Review

**DOI:** 10.3390/ijerph17228627

**Published:** 2020-11-20

**Authors:** Jia Lu, Shabana Jamani, Joseph Benjamen, Eric Agbata, Olivia Magwood, Kevin Pottie

**Affiliations:** 1Faculty of Health Sciences, McMaster University, 1280 Main St W, Hamilton, ON L8S 4L8, Canada; luj10@mcmaster.ca; 2Faculty of Medicine, University of Ottawa, 75 Laurier Ave. E, Ottawa, ON K1N 6N5, Canada; sjama085@uottawa.ca (S.J.); bjose093@uottawa.ca (J.B.); 3C.T. Lamont Primary Health Care Research Centre, Bruyère Research Institute, 85 Primrose Ave, Ottawa, ON K1R 6M1, Canada; ericagbata@gmail.com (E.A.); omagwood@bruyere.org (O.M.); 4Interdisciplinary School of Health Sciences, University of Ottawa, 75 Laurier Ave. E, Ottawa, ON K1N 6N5, Canada; 5Department of Family Medicine, School of Epidemiology, Public Health and Preventive Medicine, University of Ottawa, 75 Laurier Ave. E, Ottawa, ON K1N 6N5, Canada

**Keywords:** integrated care, global mental health, primary care, migrants, refugees, scoping review

## Abstract

Migrants are at a higher risk for common mental health problems than the general population but are less likely to seek care. To improve access, the World Health Organization (WHO) recommends the integration of mental health services into primary care. This scoping review aims to provide an overview of the types and characteristics of mental health services provided to migrants in primary care following resettlement in high-income countries. We systematically searched MEDLINE, EMBASE, PsycInfo, Global Health, and other databases from 1 January 2000 to 15 April 2020. The inclusion criteria consisted of all studies published in English, reporting mental health services and practices for refugee, asylum seeker, or undocumented migrant populations, and were conducted in primary care following resettlement in high-income countries. The search identified 1627 citations and we included 19 studies. The majority of the included studies were conducted in North America. Two randomized controlled trials (RCTs) assessed technology-assisted mental health screening, and one assessed integrating intensive psychotherapy and case management in primary care. There was a paucity of studies considering gender, children, seniors, and in European settings. More equity-focused research is required to improve primary mental health care in the context of global mental health.

## 1. Introduction

Common mental disorders, including major depressive disorder, generalized anxiety disorder, posttraumatic stress disorder (PTSD), and substance-use disorders, have been found to affect one in five adults worldwide, and are becoming increasingly prevalent [1]. By 2030, depression is likely to be the second greatest burden of disease in the world and the single highest cause of disease burden in high-income countries—which has major implications for global mental health [2]. Refugees represent a global priority population with unique mental health promotion needs [3]. Around the world, there are over 28 million refugees and it is estimated that an additional 1.44 million people will need resettlement in 2020 [4,5]. More than 646,000 refugees, particularly individuals originating from Syria, are estimated to be in need of resettlement out of host states including Egypt, Iraq, Jordan, Lebanon, and Turkey [5]. North America, Europe, and Australia are the regions that resettle the most refugees and provide local integration and health resettlement services [6]. In comparison to the general population, refugees have a higher risk for depression, anxiety, and PTSD [7,8]. This is often due to pre-migration (exposure to war, violence, torture in country of origin), migration (experiencing trauma during the migration process), and post-migration (experiencing stress and trauma due to uncertainty of their refugee status, unemployment, discrimination, and social exclusion) factors [9,10,11,12]. 

Access to common mental health services is difficult for the general population and is even more challenging for culturally diverse and marginalized migrant populations [13,14,15,16,17]. Compared to the general population, migrants are less likely to seek out care for mental health conditions [15,16,17]. This is due to a number of factors including language barriers, cultural stigmatization of mental health conditions, cultural influences on patterns of coping and seeking help, and unfamiliarity with the healthcare system in the host country [18,19,20,21]. Furthermore, even when immigrants and refugees are able to access mental health services, they may face other barriers to receiving high-quality care due to cultural influences on the presentation of symptoms and the lack of or inappropriate use of interpreters [22,23,24]. 

Each year, approximately 10 to 20 percent of migrants will consult a primary care provider for a mental health problem [25]. As the main point of access to mental health services for these patients, community health centers and primary care clinics can play an important role in the early identification and treatment of mental disorders. In 2008, the World Health Organization (WHO) and the World Organization of Family Doctors (WONCA) released a report to promote the integration of mental health services into primary care [26]. The report stated several benefits of integrating mental health services into the primary care setting, such as enhancing access to mental health services, reducing health care costs, promoting the respect of human rights, and increasing the likelihood of positive health outcomes [26]. In the WHO model, primary care for mental health is an essential part of comprehensive health care and must be supported by other levels of care to meet the full spectrum of mental health needs of the population [26]. 

The terms “integrated care” and “collaborative care” have been used by several authors to describe the integration of mental health into primary care [27,28]. Collaborative care describes practices in which health professionals work together to provide care. A useful framework offered by Blount categorizes care practices as dimensions of collaborative care based on the relationship of mental health and medical providers, and describes them as “coordinated,” “co-located,” or “integrated” [29]. The Blount framework is useful in making our descriptions of collaborative care more precise to avoid confusion and to make the comparison of programs more reliable [29]. In coordinated care, primary care and mental health services exist in different settings, and primary care providers refer patients to mental health specialists [29,30]. Routine screening for mental health problems typically occurs in the primary care setting, and primary care providers may deliver behavioral health interventions using brief algorithms [29,30]. On the other hand, co-located care occurs when medical services and mental health services are located in the same facility, and there is enhanced communication between primary care providers and mental health specialists due to shared facilities [29,30]. Lastly, integrated care requires the highest degree of collaboration and communication among primary care teams [29,30]. In integrated care, a team of primary care providers and mental health specialists typically work together to deliver medical and mental health components within one treatment plan [29,30]. It is essential to integrate mental health services into migrant primary care [18,19,20,21]. Primary care providers are amongst the most commonly consulted healthcare professionals for migrants with mental health service needs [16]. Similarly, many migrants prefer to receive care in the community due to stigma with specialized mental health services [31,32,33]. Further, due to the complex and multifaceted health needs of this population (e.g., somatic symptom presentation of mental illnesses, potential exposure to infectious diseases), primary health care clinics can be seen as the ‘medical home’ to migrant populations in which medical and mental health needs are addressed concurrently [26,33,34,35]. Although there have been some systematic reviews that supported the integration of mental health services into primary care for the general population, there has been a lack of systematic or scoping reviews summarizing the evidence on integrated services for migrant populations [36,37,38,39,40]. 

This scoping review aims to provide an overview of the characteristics and range of mental health services and practices provided to migrants in primary care following resettlement in high-income countries. In April 2020, we conducted a preliminary search for previous scoping and systematic reviews on the topic aligning to the same concept in the Joanna Briggs Institute (JBI) Database of Systematic Reviews and Implementation Reports and the Cochrane Database of Systematic Reviews and found no studies. For the purpose of this scoping review, we use ‘mental health services’ to represent behavior health services. We also use the term ‘migrants’ to refer to refugees, asylum seekers, and undocumented migrants. We recognize that these populations represent heterogeneous groups that face unique challenges in accessing mental health care following resettlement. 

## 2. Methods 

### 2.1. Protocol

We developed a protocol for this scoping review using the five-stage methodological framework proposed by Arksey and O’Malley [41], and further refined according to recommendations made by the JBI [42]. The final version of the protocol is available from the primary author upon request. 

This scoping review included the following five key stages: (1) Identifying the research question; (2) identifying relevant studies; (3) study selection; (4) charting the data; and (5) collating, summarizing, and reporting the results. We report our findings according to the Preferred Reporting Items for Systematic Reviews and Meta-Analyses Scoping Review (PRISMA-ScR) checklist [43] (See Appendix A).

### 2.2. Research Question

The review was guided by the question, “What are the characteristics and range of mental health services and practices provided to migrants in primary care settings following resettlement to a high-income country?” This scoping review aimed to map and characterize the types of mental health services and practices provided to migrants in primary care following resettlement in high-income countries and to identify research gaps in the existing literature.

### 2.3. Data Sources and Search Strategy

We developed a search strategy in consultation with an expert librarian (LB). We systematically searched five electronic databases from 1 January 2000 to 15 April 2020: MEDLINE, EMBASE, PsycInfo, Global Health, and Cumulative Index to Nursing and Allied Health Literature (CINAHL). The search strategy consisted of terms such as refugee, asylum seeker, primary health care, community health services, family physician, general practitioner, nurse practitioner, family health team, shared care model, mental health, mental disorder, post-traumatic stress disorder, anxiety disorder, and depression, and were combined using Boolean operators (see Appendix A for complete search strategy). The search query was tailored to the specific requirements of each database. We also scanned references of the included articles for any relevant studies. Companion reports were identified by matching the authors and mental health intervention and were used for Appendix A only.

### 2.4. Eligibility Criteria

We included articles that met the following criteria: (1) Included refugee, asylum seeker, or undocumented migrant populations; (2) described mental health services and practices; and (3) conducted in primary care settings following resettlement in high-income countries (see Table 1 for full inclusion criteria). Refugees, asylum seekers, and undocumented migrants were defined using the United Nations High Commissioner for Refugee’s (UNHCR) definitions. Refugee is defined as “someone who is unable or unwilling to return to their country of origin owing to a well-founded fear of being persecuted for reasons of race, religion, nationality, membership of a particular social group, or political opinion” [44]. Asylum seeker is defined as “someone whose request for sanctuary has yet to be processed” [45]. Undocumented migrants are defined as “persons who do not fulfil the requirements established by the country of destination to enter, stay, or exercise an economic activity” [46]. We included all refugees, asylum seekers, and undocumented migrants of any age and ethnicity, but immigrants, internally displaced persons, and all other populations were excluded from this scoping review. The primary care setting is defined as “the first level of care within the formal health system” [26] and is mostly led by family physicians, general practitioners, pediatricians, or nurse practitioners [47]. We excluded studies that focused on tertiary/specialist care. The essential mental health services at the primary care level include “early identification of mental disorders, treatment of common mental disorders, management of stable psychiatric patients, referral to other levels where required, attention to the mental health needs of people with physical health problems, and mental health promotion and prevention” [26]. Lastly, high-income countries were defined using the World Bank’s definition of high-income economies for the 2020 fiscal year: “Those with a Gross National Income (GNI) per capita of $12,376 or more” [48]. We excluded studies that took place in low- and middle-income countries. 

Due to time and resource constraints, we applied date and language restrictions in order to select articles that were most relevant. We excluded studies that were published before 2000 and those published in languages other than English. Conference proceedings, articles without full text, book chapters, book reviews, vignette studies, commentaries, guidelines, study protocols, and editorials were also excluded. In addition, articles that only reported on health needs, barriers, and challenges to healthcare access among refugees without presenting specific interventions and practices were also excluded. Furthermore, articles that only included training or workshops aimed to improve the competencies of primary care providers were also excluded. 

### 2.5. Study Selection Process

Search results were imported into COVIDENCE, an online systematic review software [49]. The inclusion criteria were also imported and were used for screening titles and abstracts during level 1 screening, and full-text articles during level 2 screening. 

To ensure reliability between reviewers, a series of pilot tests was conducted before title and abstract screening. During each pilot test, 20 articles were used to evaluate inter-rater agreement. Once the percent agreement reached 70%, we proceeded to the next stage. If lower agreement was observed, the inclusion and exclusion criteria were clarified, and another pilot-test occurred. Two rounds of pilot tests were required for title and abstract screening on a random sample of 40 articles in total. Subsequently, two reviewers (JL and SJ) independently screened the title and abstract of each article for inclusion. Due to time constraints, titles for which an abstract was not available were excluded. Reviewers met throughout the screening process to resolve conflicts and discussed any uncertainties that arose during the study selection process. 

All articles deemed relevant after title and abstract screening were included for full-text screening. For full-text screening, one round of pilot test was conducted on a random sample of 20 articles in total. Using the same process, groups of two reviewers (JL, JB, EA, SJ) subsequently screened the full text of potentially relevant articles to determine eligibility. Disagreements within each group were resolved through discussion between the two reviewers.

### 2.6. Data Extraction

A standardized data extraction template was developed with input from the entire review team. For all the articles included in the final analysis, data were extracted on the following variables: (1) Author and year of publication, (2) study setting and country, (3) study population (type, age, gender, country of origin, sample size), (4) study objective, (5) study design, (6) type of approach to care (categorized using Blount’s framework [29]), (7) type of professional(s) involved, (8) type of mental health service(s) offered (and type of psychotherapeutic intervention(s), if any), (9) comparator (if any), and (10) impact measures. 

Using a random sample of five included studies, the data extraction form was calibrated amongst the team. Subsequently, each included article was extracted by groups of two reviewers (JL, JB, EA, SJ). Disagreements within each group were resolved through discussion between the two reviewers. 

### 2.7. Methodological Quality Appraisal

We did not appraise the methodological quality or risk of bias of the included articles, which is consistent with guidance on scoping review conduct [42].

### 2.8. Data Summary and Synthesis

The data were compiled in a single spreadsheet and summarized quantitatively using Microsoft Excel. The frequencies were calculated for the following variables: Year of publication, study setting, study country, study design, type of participants, patient gender, patient age group, patient country of origin, type of approach to care, type of professional(s) involved, type of mental health service(s) offered, type of psychotherapeutic intervention(s) offered, and impact measure(s) reported in the intervention studies. Gaps in the literature were also identified.

## 3. Results

### 3.1. Literature Search

A total of 1627 records were identified through database searching, and five additional records were identified through scanning the reference lists of included studies. After deduplication (removal of duplicate citations), 975 records were screened by title and abstract. After screening 117 potentially relevant full-text articles, 93 articles were excluded as they did not meet the inclusion and exclusion criteria. Most articles were excluded because they were not conducted in the primary care setting, did not include mental health services or interventions, were not written in English, or were editorials or abstract proceedings. Subsequently, 19 articles were included: Asgary et al., 2006 (study 1); Bertelsen et al., 2018 (study 2); Bosson et al., 2017 (study 3); Dalgaard et al., 2020 (study 4); Dick et al., 2015 (study 5); Schaeffer and Jolles, 2019 (study 6); Ahmad et al., 2017 (study 7); Furler et al., 2010 (study 8); Jensen et al., 2013 (study 9); Kirmayer et al., 2003 (study 10); Samarasinghe et al., 2010 (study 11); McMahon et al., 2007 (study 12); Njeru et al., 2016 (study 13); Northwood et al., 2020 (study 14); Polcher and Calloway, 2016 (study 15); Rousseau et al., 2013 (study 16); Sorkin et al., 2019 (study 17); Weine et al., 2003 (study 18); White et al., 2015 (study 19) [24,50,51,52,53,54,55,56,57,58,59,60,61,62,63,64,65,66,67,68,69,70,71]. Five articles were used as companion reports [68,69,70,71,72]. The details of the search process are presented in Figure 1. Characteristics of included studies are summarized in Table 2.

### 3.2. Study Characteristics

Of the 19 studies, 10 were carried out in the United States (US) primary care (study 1–3, 6, 13–15, 17–19), three in Canada (study 7, 10, 16), two in Denmark (study 4 and 9), one in Israel (study 5), one in Australia (study 8), one in Sweden (study 11), and one in Ireland (study 12). Four studies were published in the 2000s (study 1, 10, 12, 18), 14 in the 2010s (study 2–9, 11, 13, 15–17, 19), and one in 2020 (study 14). Seven out of the 20 studies were conducted in community health centers or in a community health organization (study 6, 7, 11, 15–18), six in specialized health clinics for refugees or asylum seekers (study 1–5, 19), four in general practice or primary care clinics (study 9, 12–14), one in a cultural consultation service affiliated with a hospital (study 10), and one in both primary care clinics and a community health center (study 8). 

As this scoping review aimed to map out the types and characteristics of mental health services and practices in the primary care setting, a variety of study designs were included. Four of the studies were qualitative studies that interviewed participants (study 4, 8, 9, 11), three were randomized controlled trials (RCTs) (study 7, 14, 17), three were retrospective chart review studies (study 1, 2, 12), two were cohort studies (study 13 and 15), and two were case reports or case series (study 3 and 16). A cross-sectional study (study 5), a mixed-methods study (study 10), a quality improvement project (study 6), a feasibility study (study 18), and a quasi-experimental retrospective study (study 19) were also included.

### 3.3. Participant Characteristics

In the 19 included studies, one study included both refugees and asylum seekers (study 5), and two included refugees, asylum seekers, and undocumented migrants (study 7 and 10). On the other hand, eight studies specifically focused on refugees (study 3, 13–19), three on asylum seekers (study 1, 2, 12), and five studied professionals involved in providing health care to migrant populations (study 4, 6, 8, 9, 11). The number of participants in each study varied greatly; studies that included patients as participants ranged from 3 to 3511 while those that included health care professionals as participants ranged from 8 to 34.

Of the studies that specified the age of migrant populations, most (*n* = 12) included adults (study 1–3, 5, 7, 12–15, 17–19). Three studies included seniors (study 3, 5, 19), three included children (study 3, 12, 16), and only one studied children as the target group (study 16). Most of the studies (*n* = 13) included both men and women (study 1–3, 5, 7, 10, 12–18), while one study included women exclusively (study 19). In addition, the majority of studies (*n* = 8) were non-targeted and included participants from multiple countries of origin (study 1–3, 5, 7, 10, 12, 15, 16). Five studies used country of origin to select participants; two studies only included migrant populations from East Asia and the Pacific (study 14 and 17), one only included migrants from Europe and Central Asia (study 18), and one only included migrants from Sub-Saharan Africa (study 19).

### 3.4. Approach to Care

Several distinct approaches to care were utilized by health care professionals to work with migrants in the primary care setting. In all of the studies, health care professionals practiced cross-cultural care and communication, such as working with interpreters or culture brokers when communicating with patients and/or recognizing the influence of culture on the perception of mental disorders and their treatment. In four of the studies, health care professionals also reported to recognize the pervasive impact of trauma on migrant health and practiced trauma-informed care (study 3, 9, 17, 19). In terms of the level of integration, four studies were conducted in primary care settings with coordinated care (study 9, 10, 15, 18), in which primary care providers referred patients to mental health specialists. For example, the study by Jensen et al. described a primary care clinic in which the primary care physicians diagnosed, prescribed medications, provided psychotherapy, and referred patients to community mental health services as needed (study 9). Five studies were conducted in clinics with co-located care (study 2, 5, 14, 16, 19), in which medical services and mental health services are located in the same facility but the organizational structures are not merged. For example, the study by Northwood et al. described two urban primary care clinics in which primary care physicians, psychotherapists, and case managers provided care at the same site (study 14). While most of the organizational structure remains not merged, patients are able to take advantage of the proximity of medical, mental health, and social services (study 14). Seven studies were conducted in integrated care clinics (study 1, 3, 4, 6, 7, 13, 17), and teams of primary care providers and mental health specialists worked together to deliver medical and behavioral health components within one treatment plan. For example, the study by Dalgaard et al. described a treatment center in which social workers, physiotherapists, psychologists, and primary care physicians worked in a team to provide treatment to every family (study 4). After the initial assessment, professionals in the team work together to develop a comprehensive case formulation and treatment plan for each family (study 4).

### 3.5. Professionals Involved and Mental Health Services

The professionals involved and the range of mental health services provided varied considerably between studies. In the majority of studies (*n* = 16), a primary care physician was involved (study 1–9, 12–17, 19), while nurses were involved in six studies (study 2, 5, 10, 13, 15, 18), advanced practice registered nurses (APRNs) in four (study 3, 6, 7, 11), and medical residents in three (study 1, 10, 17). In terms of mental health specialists, seven studies included psychiatrists (study 3, 5, 7, 10, 13, 16, 18), five included psychologists and/or psychotherapists (study 3, 4, 10, 14, 19), three included mental health trainees (study 2, 3, 10), and two did not specify the type of mental health professionals involved (study 2 and 16). Seven studies included case managers and/or social workers who helped patients with accessing social services and/or provided mental health services (study 3, 4, 6, 7, 10, 14, 16). To help with cross-cultural communication, interpreters were involved in 14 studies (study 1–10, 13–15, 19) and culture brokers/global health navigators in two studies (study 3 and 10). Some of the studies also included other professionals, including physiotherapists (*n* = 2) (study 4 and 5), dieticians (*n* = 1) (study 5), acupuncturists (*n* = 1) (study 5), medical anthropologists (*n* = 1) (study 10), medical assistants (*n* = 1) (study 15), trained lay workers (*n* = 1) (study 18), and school staff (*n* = 1) (study 16). 

In most studies (*n* = 17), health care professionals diagnosed mental disorders in patients (study 1–10, 12–17, 19), while written mental health screening tools and technology-assisted screening tools were also implemented in seven (study 1–3, 5, 6, 13, 15) and two studies (study 7 and 17), respectively. Four studies used only pharmacologic management (study 2, 5, 12, 13), and four studies offered only psychotherapeutic interventions (study 3, 6, 14, 16). The other studies (*n* = 9) used a more comprehensive approach with a choice, or combination of, pharmacotherapy and psychotherapeutic interventions (study 4, 7–10, 15, 17–19). Two studies offered mental health treatment to patients but did not provide the details of treatment (study 1 and 11). In seven studies, health care professionals referred patients to community mental health services outside of the primary care team (study 2, 4, 6, 9, 15, 17, 19), while one offered brief intervention (study 6). In addition, social work and/or case management was offered in two studies (study 4 and 14), physiotherapy in one study (study 4), and recommendations to treatment were provided in one study (study 10). 

Of the studies that offered psychotherapeutic interventions, the majority (*n* = 8) offered individual counselling but did not describe the psychotherapeutic interventions in detail (study 6–9, 15, 17–19). Of those that provided such detail, four studies offered cognitive behavioral therapy (CBT) (study 3, 4, 10, 14), four offered family and couple therapy (study 3, 4, 8, 10), four provided psychoeducation (study 4, 14, 16, 18), two offered support groups (study 3 and 18), one offered narrative exposure therapy (NET) (study 14), one used motivational interviewing (study 14), and one used somatic experiencing (study 4). In the study with children as the target group (study 16), humanistic therapies, such as art therapy and play therapy, were offered.

### 3.6. Interventions and Impact Characteristics

Of the 19 studies, eight were intervention studies, which include three RCTs, two cohort studies, a quasi-experimental retrospective study, a quality improvement project, and a feasibility study. The evaluated interventions varied considerably between studies. Four studies evaluated the implementation of written or technology-assisted mental health screening tools, often in combination with other interventions (study 6, 7, 15, 17). The studies reported on the impact of the intervention on screening rates (*n* = 2) (study 6 and 15), diagnosis rates (*n* = 4) (study 6, 7, 15, 17), and access to community mental health treatment (*n* = 4) (study 6, 7, 15, 17). In addition, two studies reported its impact on clinical outcomes for patients (study 6 and 17), one on the discussion of mental health issues during primary care consultation (study 7), and one on patient adherence to treatment regime (study 6). Three studies evaluated the implementation of collaborative care management, in which primary care providers worked with mental health professionals in the same facility to provide mental health care to patients (study 13, 14, 19). Reported impact measures differed between the studies, and included diagnosis rates (study 14), access and/or utilization of community mental health treatment (study 13), patient adherence to treatment regime (study 19), clinical outcomes for patients (study 14), healthcare utilization (study 19), social functioning outcomes (study 14), and feasibility of the model (study 13). Lastly, one study evaluated a multi-family support group intervention (study 18), and reported its impact on psychiatric service utilization, perception of social supports, and mental health knowledge. 

## 4. Discussion

This scoping review identified 19 studies that systematically report on the type and characteristics of mental health services and practices provided to migrants in primary care in high-income countries. Our analysis of these studies shows promising programs and also identifies research gaps in the existing literature. There is a paucity of intervention studies on the topic, especially considering the growing nature of forcibly displaced migrants [6]. This paucity may be due to the ethical, methodological, and resource-related challenges of conducting experimental research with migrant populations [73,74]. As a result of these challenges, research studies in the field often have small sample sizes, a lack of control groups, and lack of randomization [73,74]. Based on our analysis of the existing evidence base, we have identified a number of research gaps that require further investigation. 

The majority of studies were conducted in the US. Although the US hosts a sizable population of forcibly displaced migrants, this significantly underrepresents countries that host significantly more forcibly displaced migrants. For example, Germany resettled almost 1.5 million refugees and asylum seekers, the third largest number in 2019 [6]. More research is needed on mental health services for refugees in primary care outside of the US. In addition, this scoping review was limited to studies in high-income countries, thus excluding studies conducted in countries such as Turkey, Colombia, and Pakistan. Considering that 85% of forcibly displaced migrants are hosted in developing countries, more research is needed on low- and middle-income countries [6]. Certainly, mental health services in primary care may be different and predominantly delivered by nurses or trained lay workers in low- and middle-income countries [26].

Most studies included adult patients, while few focused specifically on children and seniors. In addition, for migrant status, most studies included refugees, while none of the studies specifically focused on undocumented migrants and unaccompanied minors. By comparison, 40% of forcibly displaced persons were children below 18 years of age in 2019, and around 400,000 unaccompanied and separated children sought asylum between 2010 and 2019 [6]. Taking all these characteristics into account, future studies should better reflect the demographic profiles of migrant populations, particularly children and unaccompanied minors.

All studies reported that cross-cultural care and communication were practiced by health care providers. By comparison, trauma-informed care was reported to be practiced by providers in only four studies. Trauma-informed care is an approach that recognizes the pervasive impact of trauma [75], and has been identified as best practice for the care of migrant youth by both the American Academy of Pediatrics and the Budapest Declaration on the Rights, Health, and Well-Being of Children and Youth on the Move [76,77]. However, many primary care providers do not discuss trauma histories with patients or feel unprepared to do so [78,79]. Future studies should assess trauma-informed care for primary care providers [65,80]. For vulnerable populations, socioeconomic stressors are closely connected to their mental health and can prevent them from benefiting from mental health treatment [54]. The study by Dalgaard et al. [54] found that it was important to help parents determine the most immediate problems in order to stabilize the family and help them benefit from therapy; we identified two studies that offered social services and case management to patients [54,60]. 

We identified only three RCTs; two on technology-assisted mental health screening and one on integrating intensive psychotherapy and case management (IPCM) into primary care. In the study by Ahmad et al. [50], an interactive computer-assisted client assessment survey (iCCAS) tool improved mental health consultations compared to usual care. Similarly, the study by Sorkin et al. [65] found that a multicomponent health information technology mental health screening intervention helped primary care providers with diagnosing and providing evidence-informed care to Cambodian refugees. In terms of integrating mental health care into primary care, the study by Northwood et al. [60] conducted a pragmatic RCT on adult Karen refugees in two primary care clinics. Compared to baseline, IPCM patients showed significant improvements in depression, PTSD, anxiety, and pain symptoms and in social functioning at 3, 6, and 12 months. By comparison, care-as-usual patients did not show significant improvements. More experiments using good between-group design is required to empirically validate the intervention [81]. Future studies are required with more heterogeneous groups of migrants to examine the effectiveness of the intervention more broadly. Furthermore, cost effectiveness analyses are needed to demonstrate the benefit of co-located and integrated care to the healthcare system.

A range of the psychotherapeutic interventions were offered in the studies, including CBT, family and couple therapy, support groups, and NET. Although psychotherapeutic interventions were offered, none of the studies evaluated the effectiveness of psychotherapeutic interventions for migrants in primary care. 

Due to the lack of studies, the effectiveness of mental health interventions in primary care has not been established with migrant populations. Although the primary care mental health interventions that have been validated for the general population may be valid for migrant populations, as scientists, we cannot with certainty assume this indirect evidence will apply to refugees [81]. Furthermore, these studies may further validate the importance of considering cultural factors in treatment. 

### Strengths and Limitations of this Scoping Review

A strength of this scoping review is that we conducted this review using a predefined protocol that followed Arksey and O’Malley’s framework and the JBI guidance [41,42]. In addition, the studies were carefully selected based on a set of predefined eligibility criteria and have yielded potentially useful information regarding the type of mental health interventions that have been evaluated for migrants in primary care. However, there are several limitations in this review that need to be considered.

Due to the scoping approach of this review, we did not appraise the quality of the evidence and should be cautious in reporting the impact of each intervention. The scoping review was also limited to published peer-reviewed studies and studies published in English. As such, it is likely that other relevant information was not captured, for example, books, grey literature, and publications from non-English-speaking countries. A further limitation is the variation in locations included as ‘primary care’ and the broad concept of integrated care. Many studies lacked details of the study setting and did not specify the relationship between mental health and medical providers. 

## 5. Conclusions

This scoping review has highlighted the current evidence on mental health interventions in primary care targeted at migrant populations in high-income countries and identified research gaps in the existing literature. However, the studies are not globally representative, and thus, approaches and interventions may not be generalizable. Future research should include evidence from low- and middle-income countries. More research is also needed to validate the integrated primary and mental health care model for migrant populations, disentangle elements that make integrated care effective for the population, and evaluate the cost-effectiveness of the integrated care model for the population. Future research teams should consider applying systematic review methods to determine the effectiveness of different care practices on clinical and health system outcomes. In addition, future research studies should consider PROGRESS equity factors and pediatric populations to evaluate the equity effectiveness of psychotherapeutic interventions in primary care for migrant populations. PROGRESS equity factors include place of residence, race/ethnicity/language, occupation, gender/sex, religion, education, socioeconomic status, and social capital [82]. Better reporting of the study setting and the relationship between mental health and medical providers should also be encouraged in future studies.

## Figures and Tables

**Figure 1 ijerph-17-08627-f001:**
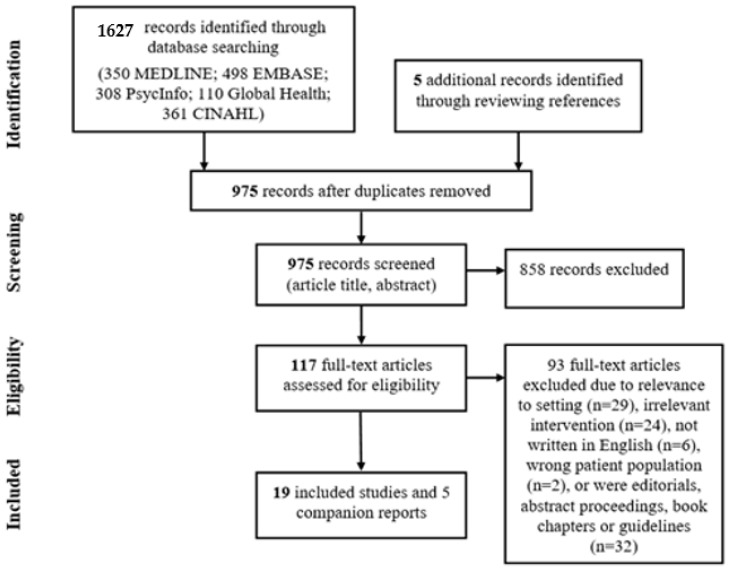
The Preferred Reporting Items for Systematic Reviews and Meta-Analyses (PRISMA) study flow diagram.

**Table 1 ijerph-17-08627-t001:** Inclusion criteria.

Inclusion Criteria	Definition	
Population	Refugee	“Someone who is unable or unwilling to return to their country of origin owing to a well-founded fear of being persecuted for reasons of race, religion, nationality, membership of a particular social group, or political opinion” [44]
	Asylum seeker	“Someone whose request for sanctuary has yet to be processed” [45]
	Undocumented migrant	“Persons who do not fulfil the requirements established by the country of destination to enter, stay, or exercise an economic activity” [46]
Concept	Mental health services: “Early identification of mental disorders, treatment of common mental disorders, management of stable psychiatric patients, referral to other levels where required, attention to the mental health needs of people with physical health problems, and mental health promotion and prevention” [26]	
Context	1. Primary care setting: “The first level of care within the formal health system” [26] and is mostly led by family physicians, general practitioners, pediatricians, or nurse practitioners [47]	
	2. High-income countries: “Those with a Gross National Income (GNI) per capita of $12,376 or more” [48]	

**Table 2 ijerph-17-08627-t002:** Characteristics of included studies.

Author(s)/Year	Design	Setting	Participant Description	Care Approach	Professionals Involved	Mental Health Services and Intervention	Collaboration between Professionals	Intervention	Comparison
Asgary et al. 2006	Retrospective chart review	A human rights clinic affiliated with a medical center (US)	Adult asylum seekers from multiple countries of origin	Cross-cultural care and communication; integrated care	PCPs, medical residents, interpreters	Screening, diagnosis, unspecified treatment	Medical residents screened, diagnosed, and treated patients, while precepted by the attending physicians.	N/A	N/A
Bertelsen et al. 2018	Retrospective chart review	A program for survivors of torture affiliated with a medical center (US)	Adult asylum seekers from multiple countries of origin	Cross-cultural care and communication; co-located care	PCPs, nurses, mental health professionals (unspecified), trainees of mental health professionals, interpreters	Screening, diagnosis, pharmacotherapy, referral to community mental health services	Mental health professionals or their trainees screened, diagnosed, and referred patients to appropriate services. PCPs provided pharmacologic management.	N/A	N/A
Bosson et al. 2017	Case report	A global health center affiliated with a university (US)	Refugee children, adults, and seniors from multiple countries of origin	Cross-cultural care and communication; integrated care; trauma-informed care	PCPs, APRNs, psychologists/psychotherapists, psychiatrists, social workers/case managers, trainees of mental health professionals, interpreters, global health navigators	Screening, diagnosis, psychotherapy (CBT, family and couple therapy), facilitation of support groups	Psychologists and PCPs screened and diagnosed patients, while psychologists and psychiatrists provided psychotherapy. Trainees of mental health professionals were also included to assess and treat patients. Global health navigators assisted with interpreting, advocacy, and connecting with local groups and agencies.	N/A	N/A
Dalgaard et al. 2019	Qualitative	A treatment center for torture victims (Denmark)	Health care providers	Cross-cultural care and communication; integrated care	PCPs, psychologists/psychotherapists, social workers/case managers, interpreters, physiotherapists	Diagnosis, pharmacotherapy, psychotherapy (CBT, family and couple therapy, psychoeducation, somatic experiencing), referral to community mental health services, social work services, physiotherapy	PCPs, psychologists, social workers, and physiotherapists all initially assessed patients. Psychologists and physiotherapists provide psychotherapy, while GPs provide pharmacologic management and refer complex cases to psychiatrists. Physiotherapists provide individual physiotherapy. Social workers help families navigate the Danish system for social services.	N/A	N/A
Dick et al. 2015 Companion article: Lurie 2009	Cross-sectional	A human rights clinic (Israel)	Adult and elderly refugees, asylum seekers, and undocumented migrants from multiple countries of origin	Cross-cultural care and communication; co-located care	PCPs, nurses, psychiatrists, interpreters, physiotherapist, dietitians, acupuncturists	Screening, diagnosis, pharmacotherapy	PCPs screened and diagnosed patients, while psychiatrists prescribed medications.	N/A	N/A
Schaeffer et al. 2019	Quality improvement project	A community health center (US)	Health care providers	Cross-cultural care and communication; integrated care	PCPs, APRNs, social workers/case managers, interpreters	Screening, diagnosis, psychotherapy, referral to community mental health services, brief intervention	PCPs and APRNs screened, diagnosed, offered brief intervention (Option Grid), and referred patients to community mental health services. Clinical social workers provided psychotherapy.	Use of written standardized Patient Health Questionnaire (PHQ) screening tools in six languages, the Option Grid for clients who screen positive for depression, a “right care” tracking log for screen- positive clients, and team meetings to support capacity building.	The same community health center before the implementation of the quality improvement project.
Ahmad et al. 2017 Companion articles: Ahmad et al. 2016 Ferrari et al. 2016 Ferrari et al. 2018	RCT	A community health center (Canada)	Adult refugees and asylum seekers from multiple countries of origin	Cross-cultural care and communication; integrated care	PCPs, APRNs, psychiatrists, social workers/case managers, interpreters	Technology-assisted screening, diagnosis, pharmacotherapy, psychotherapy	Research assistants screened patients for common mental disorders using an interactive survey on an iPad. PCPs diagnosed, prescribed medications, and referred patients to other mental health professionals. Social workers offered psychotherapy, while psychiatrists provided pharmacologic management.	An Interactive Computer-Assisted Client Assessment Survey (iCCAS) that is used to detect common mental disorders.	Care as usual with no health-risk assessments before the consultation
Furler et al. 2010	Qualitative	A community health center and primary care clinics (Australia)	Health care providers	Cross-cultural care and communication	PCPs, interpreters	Diagnosis, pharmacotherapy, psychotherapy (family and couple therapy, unspecified individual counselling)	PCPs diagnosed, prescribed medications, and provided psychotherapy to patients.	N/A	N/A
Jensen et al. 2013	Qualitative	Primary care clinics (Denmark)	Health care providers	Cross-cultural care and communication; coordinated care; trauma-informed care	PCPs, interpreters	Diagnosis, pharmacotherapy, psychotherapy, referral to community mental health services	PCPs diagnosed, prescribed medications, provided psychotherapy, and referred patients to community mental health services.	N/A	N/A
Kirmayer et al. 2003	Mixed methods (qualitative and quantitative methods)	A cultural consultation service affiliated with a hospital	Refugees and asylum seekers from multiple countries of origin	Cross-cultural care and communication; coordinated care	Medical residents, psychiatric nurses, psychologists/psychotherapists, psychiatrists, social workers/case managers, trainees of mental health professionals, interpreters, culture brokers, medical anthropologists	Diagnosis, pharmacotherapy, psychotherapy (CBT, family and couple therapy), provide recommendations to treatment	Mental health professionals diagnosed patients, provided pharmacotherapy, psychotherapy, and recommendations to treatment.	N/A	N/A
Samarasinghe et al. 2010	Qualitative	Primary health centers (Sweden)	Health care providers	Cross-cultural care and communication	APRNs	Health promotion	APRNs engaged in health promotion.	N/A	N/A
McMahon et al. 2007	Retrospective chart review	Two primary care clinics (Ireland)	Asylum seekers (children and adults) from multiple countries of origin	Cross-cultural care and communication	PCPs	Diagnosis, pharmacotherapy	PCPs diagnosed patients, provided pharmacotherapy, and referred patients to other services.	N/A	Irish citizens
Njeru et al. 2016	Retrospective cohort	Primary care clinics affiliated with a medical center (US)	Adult refugees	Cross-cultural care and communication; integrated care	PCPs, nurses, psychiatrists, interpreters	Screening, diagnosis, pharmacotherapy	Psychiatrists provided oversight, while PCPs screened, diagnosed, and prescribed medications. Nurses served as the care manager and interacted with patients through face-to-face and telephone visits.	Collaborative care management (CCM) model	N/A
Northwood et al. 2020 Companion article: Esala et al. 2018	RCT	Two urban primary care clinics (US)	Adult refugees from East Asia and the Pacific	Cross-cultural care and communication; co-located care; trauma-informed care	PCPs, nurses, psychologists/psychotherapists, social workers/case managers, interpreters	Diagnosis, pharmacotherapy, psychotherapy (CBT, psychoeducation, motivational interviewing, NET), case management, social work services	PCPs diagnosed and prescribed medication. Psychotherapists offered psychotherapy, and case managers/social workers helped with case management and social work services. Psychotherapists and case managers did not write prescriptions; however, they flagged medical issues noted by participants for PCPs.	Intensive, coordinated psychotherapy and case management	Care as usual from PCPs (behavioral health referrals and/or brief onsite interventions)
Polcher et al. 2016	Cohort	A community health clinic (US)	Adult refugees from multiple countries of origin	Cross-cultural care and communication; coordinated care	PCPs, nurses, interpreters, medical assistants	Screening, diagnosis, pharmacotherapy, psychotherapy, referral to community mental health services	Interpreters and medical assistants screened patients, while nurses followed up and discussed the results of the screening with patients. PCPs diagnosed, prescribed medications, offered counselling, and referred patients to community mental health services when needed.	Mental health screening using the Refugee Health Screener–15	N/A
Rousseau et al. 2013	Case series	Community-based health and social services clinics (Canada)	Refugee children from multiple countries of origin	Cross-cultural care and communication; co-located care	PCPs, psychiatrists, social workers/case managers, mental health professionals (unspecified), school staff	Diagnosis, psychotherapy (humanistic therapies, psychoeducation)	The youth mental health team in the community-based health and social services clinics diagnosed and provided psychotherapy to patients, while the child psychiatry cultural consultants offered cultural consultation.	N/A	N/A
Sorkin et al. 2019	RCT	Two community health centers (US)	Adult refugees from East Asia and the Pacific	Cross-cultural care and communication; integrated care; trauma-informed care	PCPs, medical residents	Technology-assisted screening, diagnosis, pharmacotherapy, psychotherapy, referral to community mental health services	Research assistant screened patients for depression and PTSD using an iPad that administered the screening tools in patients’ preferred language. PCPs diagnosed patients, provided pharmacologic management and psychotherapy, and referred patients to community mental health services.	First, PCPs completed an online tutorial on how to provide culturally competent, trauma-informed mental health care to the Southeast Asian population. The second component involved screening all patients just before their appointment using an iPad that administered the screening tools. The third component involved giving PCPs access to evidence-based clinical algorithms and guidelines through a web-based mobile application.	Minimal intervention control condition
Weine et al. 2003	Feasibility study	A community health organization (US)	Adult refugees from Europe and Central Asia	Cross-cultural care and communication; coordinated care	Lay workers, nurses, psychiatrists	Pharmacotherapy, psychotherapy (psychoeducation, support groups)	Lay workers are trained to provide family outreach and multi-family group sessions, while an outreach team of a psychiatrist and nurse from their partnering clinic provided psychotherapy or medications for refugees requesting or needing those services.	Family outreach and multi-family group sessions.	N/A
White et al. 2015	Quasi-experimental retrospective	A Somali primary care clinic affiliated with a medical center (US)	Female refugees (adults and seniors) from Sub-Saharan Africa	Cross-cultural care and communication; co-located care; trauma-informed care	PCPs, psychologists/psychotherapists, interpreters	Diagnosis, pharmacotherapy, psychotherapy, referral to community mental health services	PCPs provided a four-visit staged approach to trauma assessment in office, including diagnosis, pharmacologic management, and referral to psychologists, while psychologists provided psychotherapy.	Staged but flexible four-visit protocol for addressing physical and psychological complaints by the PCPs, trauma-informed psychotherapy provided by psychologists, as well as co-management of patients receiving physical and mental health services.	2 post-hoc groups (therapy adherents and therapy non-adherents)

PCP = primary care physician; APRN = advanced practice registered nurse; RCT = Randomized controlled trial; CBT = cognitive behavioural therapy; NET = narrative exposure therapy.

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
