# Peer review of "Global Mental Health and Services for Migrants in Primary Care Settings in High-Income Countries: A Scoping Review"

_ijerph, 2020, doi:10.3390/ijerph17228627_

Round 1
Reviewer 1 Report
This study used the information obtained from several databases to review primary mental health services and practices pertaining to refugees as described by the United Nations High Commissioner for Refugees definition. The literature review focused on common mental health disorders found such as depression, anxiety, PTSD and substance abuse and found that refugees have a higher risk due to pre-migration markers. The authors clearly define concepts in the document. Finally, the study highlighted the importance of mental health workers in primary health care. This is important for global policies
Protocols were outlined in five stages and the research question, data sources and search strategy meticulously stated. Selection and review criteria were stipulated. The study addresses important global issues regarding refugees and identified gaps (in particular amongst children) in research studies.
Figures and Tables are pertinent and necessary for the study. Referencing is consistent with journal style.
The article borders on technical perfection and the authors are commended for this. It was a challenge finding anything wrong with this manuscript.
Major points.
- No major points of concern.
Minor points
- Line 200 – “deduplication” I am unfamiliar with the word and merely pointing out that it may be a typing error.
- Consider tabling the findings under the topic “Results”
Reviewer 2 Report
I am grateful for the work done by the authors. It is a methodical and careful paper. However, I ask for small modifications to be made for the publication of the text.
1.- The introduction is clear and concise. Perhaps it could be added, if available, information on migration by continent. Knowing the percentages of migration reception and the health system found in those countries can give strength to the theoretical framework.
2.- Method. This is a well-developed section. I thank the authors for the trouble they took to provide a complete protocol as well as supplementary tables that clarified the whole process. I only propose two questions. It is necessary to include in the text the exclusion criteria which are as important as the inclusion criteria. On the other hand, although the authors point out the search words, it is necessary to explain the sequence of use of these words and whether Boolean actions were used. Finally, they should remember that if more than one author carried out the search, providing an index of agreement between researchers strengthens the result of the search.
3.- Results.
Although this is a purely qualitative section, all the results are clearly expressed. I only recommend that in the flowchart, the authors indicate the reasons why the articles eliminated from the research were excluded. I also recommend including a table showing the studies included and their characteristics. Visual support is important for reading the results. This should appear in the text and not as supplementary material.
4.- Discussion and conclusions.
Mainly, the conclusions section should be extended and should deal with the limitations of the study and foresight in a broader way. It should be borne in mind that the studies included are not representative of the whole world and that the results obtained can only be extrapolated to the geographical regions that appear. Other important areas such as the Mediterranean areas, for example, should be taken into account in the future. In general, it is a good review article which, in order to be published, should make clear in the conclusions its level of generalisation and limitation.
Reviewer 3 Report
The paper is interesting and I found it well developed. However, I have some comments:
Introduction:
The authors report the meaning of the term “collaborative care”: this term has not been introduced before.
The framework by Blount might be better introduced and its usefulness in the light of the study aims and of the following paper development should be reported.
Method:
The methodology section appears detailed and well framed. Authors could explain the features of the expert librarian involved in the study. Why was this person selected?
Results:
The presence of tables and graph would help the comprehension of the results.
Section 3.4. “Approach to care” could be deepened in its content, by reporting more in details the differences in approaches.
Section 3.6 “Interventions and Impact Characteristics” might explore more in depth the impact of the interventions: the type of impact is reported, while there is no discussion of the dimension and direction of the impact (e.g. lines 316-318: “Lastly, one study evaluated a multi-family support group intervention (study 18), and reported its impact on psychiatric service utilization, perception of social supports, and mental health knowledge”. How did this intervention affect psychiatric service utilization, perception of social supports, and mental health knowledge?). It would be interesting if authors could include it and discuss the interventions that better impact on the population object of study.
Discussion and Conclusions
Following my previous comment, a discussion of the impact of interventions could add value to the study.
The inclusion of theoretical and practical contribution of the study would also be valuable.
Minor issues:
Please, ensure that all the acronyms in the abstract and n the text are explicated (e.g. RCT in the abstract and JBI in the text).
The authors should deepen what they refer as “PROGRESS equity factors”.
